# Post-traumatic growth experience with kinship hematopoietic stem cells transplantation in patients with aplastic anemia: A qualitative study

Xinrui Huang[1], Yan Xu[2], Yue Pan[3], Menghua Ye[4], Ting Liu[2], Min Xu[5]*

1 Nursing Department, The First Affiliated Hospital of Zhejiang Chinese Medical University, Hangzhou, Zhejiang, China, 2 The School of Nursing, Zhejiang Chinese Medical University, Hangzhou, Zhejiang, China, 3 School of Nursing and Health, Zhejiang Changzheng Vocational and Technical College, Hangzhou, Zhejiang, China, 4 Department of Hematology, The First Affiliated Hospital of Zhejiang Chinese Medical University, Hangzhou, Zhejiang, China, 5 Dean's office, The First Affiliated Hospital of Zhejiang Chinese Medical University, Hangzhou, Zhejiang, China

* yudi1212@163.com

## Abstract

### Background

A Hematopoietic stem cell transplant is the only way to cure aplastic anemia. Selection of relative donors can improve the chance of transplantation. However, the treatment of kinship transplantation and disease both bring traumatic experience to patients. Understanding the specific psychological changes in a patient's post-traumatic growth experience can help improve the quality of care.

### Objective

The purpose of this study is to explore the post-traumatic growth experience in patients with aplastic anemia received kinship hematopoietic stem cell transplantation.

### Methods

A homogeneous small-sample strategy to select aplastic anemia kinship hematopoietic stem cell transplantation patients and medical staff in the Department of Hematology of a tertiary hospital in China as study participants. Face-to-face semi-structured interview method was used to collect verbal and nonverbal data. Data analysis started from a single case, and after completing the detailed analysis of the first text, the next text was followed until sufficiently rich research information was obtained. Interpretative phenomenological analysis was used to collect verbal and nonverbal data through purposive sampling.

**Data availability statement:** All relevant data are within the manuscript and its Supporting information files.

**Funding:** This work was funded by the Zhejiang Health Science and Technology Project. The Project number is 2023KY871. The funding support for this study comes entirely from the project and does not require any additional funding from the researcher's institution. The corresponding author Xu Min is the recipient of this project and played a role in study design, decision to publish, and preparation of the manuscript.

**Competing interests:** The authors have declared that no competing interests exist.

## Results

By analyzing the interview data on the post-traumatic growth experiences of six aplastic anemia kinship hematopoietic stem cell transplant patients and three medical staff, four themes and nine sub-themes were developed.

## Conclusions

This study reveals the specific process of posttraumatic growth in patients with aplastic anemia kinship hematopoietic stem cell transplantation. Clinical workers can provide targeted psychological interventions based on the stages of patients' post-traumatic growth in order to improve the status of post-traumatic growth and quality of life of patients with aplastic anemia kinship hematopoietic stem cell transplantation.

## Introduction

Aplastic anemia (AA) is a comprehensive disease of bone marrow hematopoietic failure, which is mainly characterized by thrombocytopenia of whole blood cells and bone marrow cytopenias, and clinically manifested by anemia, hemorrhage, and infection [1]. The annual incidence of remyelination in Europe and the United States is about 2–3/1 million. The incidence rate in Asia, especially in East Asia, is about 2–3 times higher than that in Western countries [2]. In China, the annual incidence rate of remodeling is about 0.74/100,000 [3], and in recent years, with the aggravation of environmental pollution, the incidence rate shows an increasing trend year by year [4]. Hematopoietic stem cell transplantation (HSCT) is the only means of eradicating tumor cells and abnormal clonal cells in the body by pre-treating patients with high-dose radiotherapy and chemotherapy, and then infusing normal stem cells back into the body to rebuild normal hematopoietic and immune functions [5], and it is the only means of eradicating bone marrow failure in remyelination at present.

Due to the limitations of the lack of capacity of the Chinese Marrow Bank and the small number of hematopoietic stem cell transplantation donors, it is difficult and time-consuming to find donors with full HLA matching, and the success rate of the matching is only 1/100,000–1/50,000 [6], and many high-risk patients in need of immediate treatment have lost their chances of cure in the process of waiting. Kinship transplantation provides more opportunities for patients with AA [7]. But it may cause more serious problems for them than for unrelated transplant patients. Physiologically, patients with AA who received a kinship haplo-HSCT have a higher incidence of post-transplantation grade II-IV acute GVHD, chronic GVHD, cytomegalovirusmia, and EB viremia due to lower HLA mating sites [8–10]. In addition, a study [11] showed that patients receiving kinship haplo-HSCT may have a more pronounced decrease in grip strength, knee extensor strength, and higher fatigue after transplantation due to the use of higher doses of medications, such as corticosteroids, to counteract the rejection reaction due to the incompatibility of the locus [12]. Psychologically, compared with patients receiving unrelated transplants, patients receiving kinship transplants have a higher incidence of negative psychological problems such

as anxiety and depression due to the need to compromise the health of their family members in order to continue their own lives, and the fear of the impact of the transplantation act on the donor's quality of life in the future [13]. Socially, the act of kinship donation is more likely to be abducted by emotional pressure and moral obligations from family, friends, and society [14], making HSCT an obligation between relatives [15]. Similarly, although the act of relatives donating hematopoietic stem cells follows the voluntariness, it is inevitable that there will be some ethical problems, and due to the special nature of the kinship relationship, it leads to a certain degree of psychological burden on the patient. For patients, kinship transplantation is undoubtedly a traumatic event [16].

In recent years, with the rise of positive psychology, scholars have been expanding and transforming their research perspectives, shifting the focus from the people's problematic behaviors to exploring people's growth and development, and fully exploring people's positive qualities and psychological motivation [17]. They found that after experiencing traumatic events, in addition to helping alleviate negative psychological experiences, individuals also experience a series of positive psychological changes, such as post-traumatic growth (PTG) [18]. PTG refers to the sense and growth experienced by individuals in the process of struggling with traumatic events and situations (e.g., major illnesses, accidents, etc.) [19–20]. The production of PTG in patients helps to alleviate negative psychology [21] and enables them to better face the challenges of crisis, and patients with higher levels of PTG have higher positive emotions and psychological resilience [22]. Amouzegar et al. think that [10] patients experienced positive growth after kindred transplantation, including improved inter-family relationships and resolution of family problems, and stronger intimate relationships with family and friends. In addition, some researchers used qualitative research methods to explore the postoperative psychological experiences of related living kidney transplant patients, and the results refined four themes: PTG, remodeling the meaning of life, feeling gratitude, and coexisting with hope and challenges. It was found that patients' self-management and psychological adjustment abilities improved after kinship kidney transplantation, they were more able to appreciate the strengths and advantages of others, prioritized the order of physical health, and lived positively in the present and were full of hope for the future [23]. Patients also have an increased sense of personal responsibility and dependence after receiving a kindred transplant, and are grateful to their kindred donors and healthcare professionals, further deepening the bond between family members [24]. Similarly, PTG may exist in patients with remodeling after receiving a kindred transplant, but the exact process of PTG in patients with remodeling kindred transplantation is unclear.

Some current studies have explored the psychological experience of HSCT patients through qualitative research methods, but have not delved into the process of transitioning from trauma to growth from the perspective of AA patients after receiving a relative transplant. Compared with malignant blood diseases, as a chronic non-malignant blood disease, it is more difficult to draw the attention of the social population to this disease, and the urgent need for relatives to donate hematopoietic stem cells is more difficult to gain the recognition and understanding of a wide range of people in the society, and the patients tend to experience a higher degree of psychological trauma. Therefore, in this study, Interpretative phenomenological analysis (IPA) was used as a research method to explore the process of PTG changes and its influencing factors in patients with AA, and to propose intervention strategies to promote the psychological recovery of remodeled graft patients, aiming to further improve their quality of life.

## Materials and methods

### Study design

In this study, IPA was used as a research method to explore the PTG experience of patients with AA received kinship transplantation. IPA was formally proposed by Jonathan Smith [25] in Psychology and Health in 1996, in recent years, it has been widely used in the disciplines of clinical and counseling psychology, social psychology, and educational psychology [26]. Based on the theoretical foundations of Phenomenology, Hermeneutics, and Idiography [27], IPA is a method of phenomenological analysis that must examine an individual's lifeworld, exploring one's subjective perception and description of an object or event, and enabling the participants' life experiences to be expressed in their own way as

much as possible, rather than attempting to find objective descriptions or pre-determined categorization systems [28]. IPA focuses on case study research, and provides a special analysis of certain individuals based on the personal level, which is characterized by the individual's specificity, emphasizing the depth of the analysis, which must be comprehensive and thorough, and the specific context, emphasizing a population-specific viewpoint. It also takes into account its specific context, emphasizing the understanding of individual experiences from the perspective of a specific group of people [29–30]. Given the complexity of most human phenomena, IPA typically focuses on small samples of cases, spending a great deal of time on careful analysis of individual manuscripts within reasonably homogeneous samples, with the primary goal of gaining a deeper understanding of certain phenomena rather than categorically proposing common understandings. The research process of this study follows the Standards for reporting qualitative research (SRQR) checklist.

## Participants and setting

In this study, we used purposive sampling and homogeneous small sample strategy sampling. According to the actual situation, we selected patients with basically the same disease severity, HLA compatibility and other influencing factors, and selected patients who were more likely to provide rich information and better verbal ability. The validity and significance of the IPA study are related to the richness of the information described by the selected subjects as well as the size of the researcher's observation and analysis ability. The researcher's ability to observe and analyze, the size of the sample does not affect the results of the study, and information saturation is not a criterion for judging the appropriateness of IPA samples [31]. IPA studies tend to use small samples, with about 1–30 being acceptable.

In this study, patients in the hematology department of a tertiary hospital in Zhejiang Province of China were selected as study participants. Inclusion criteria: ① met the diagnostic criteria for AA in *the Diagnostic and Therapeutic Criteria for Hematological Diseases*; ② underwent a kinship transplantation procedure with the donor as a direct or collateral relative; ③ experienced positive changes brought by the event of a kinship transplantation; ④ ≥ 18 years old; ⑤ was free of psychiatric disorders and had normal communication skills; ⑥ voluntarily participated in the present study and signed the Informed Consent. Exclusion criteria: ① patients with confidentiality of medical information or related transplant donors; ② patients with other major life events in one year before and after HSCT, such as the death of relatives, car accidents, etc.; ③ those who refused to be recorded. The inclusion and exclusion criteria for healthcare workers are as follows. Inclusion criteria: ① Bachelor's degree or above; ② Have worked in a hematopoietic stem cell transplantation center for at least 3 years; ③ Willing to participate in this study; ④ I have a better understanding of the psychological changes in transplant patients with disabilities. Exclusion criteria: ① Poor language expression ability; ② Suffering from psychological disorders.

## Ethical considerations

This study was reviewed by the Medical Ethics Committee of the First Affiliated Hospital of Zhejiang Chinese Medical University (No. 2021-KL-057–02, S1 Fig.). Patients were recruited from January to March of the 2023. When inviting the study participants, the basic principle of informed consent was followed, and the participants were informed of the purpose and significance of the study and their right to refuse to participate or to withdraw from the interview in the middle of the study, and that they would not be treated unfairly as a result. At the same time, research participants were informed that the interview process needed to be audio-recorded, and the audio recordings were transcribed into text form by the researcher herself and de-privatized, e.g., research participants were represented by S, and numbers were used to differentiate between the research participants, e.g., S1, S2. In addition, information about the names of the research participants, the organizations they worked in, or the names of any person or place mentioned was used, such as "XX" to indicate that the research participants were identified on the premise that they were informed and voluntary. Finally, the interviews and related materials obtained for this study will be stored only on the researcher's private computer with encryption, and paper materials will be kept in a locked drawer, and all materials will be used exclusively for the purpose of this study and will be deleted and destroyed 5 years after the end of the study.

## Data collection

Face-to-face semi-structured interviews were used as the data collection method for this study. During the interviews, the movements and key information of the participants were handwritten down and recorded as interview notes. At the end of the interviews, the interviewer writes interview notes and reflection notes to support the collection and organization of textual data.

## Data analysis

According to Smith et al. [25–26], IPA analysis begins with a single case and completes a detailed analysis of the first text before moving on to the next text. The specific steps of analysis are as follows:

① **Repeated reading of transcribed texts.**  At the beginning of the analysis, researchers immerse themselves in the original data, which is the most important step for them to enter the personal world of the research participants, repeatedly reading until they have a sense of the data as a whole.

② **Preliminary annotation and critique.**  According to different focuses, the preliminary annotation and evaluation is divided into three unrelated processes as follows: a. descriptive evaluation: focusing on describing what the research participants said, i.e., the subjective dialogues in the text; b. linguistic evaluation: focusing on exploring the application of the research participants' particular language; c. conceptual evaluation: focusing on more abstract conceptualization. For specific preliminary annotations and analysis examples, please refer to S6 Table.

③ **Developing themes.**  After completing the initial critique, the researcher further analyzed the initial annotations of the research participant's transcribed text to find the themes expressed by naming the events, objects, and behaviors of the research participant in the particular context of the material. For specific examples of basic themes, please refer to S7 Table.

④ **Finding inter-theme associations.**  The list of themes was made according to the order in which the themes were extracted, and the related themes were clustered to form theme clusters after finding the correlation between the themes. The process of analysis is not subjective to the researcher's point of view, to ensure that each theme is derived from the transcribed text, and to ignore the themes that are not highlighted in the text. For specific thematic analysis examples, please refer to S8 Table.

⑤ **Focusing on the next case analysis.**  The researcher analyzed each case individually

⑥ **Finding thematic patterns between cases.**  The thematic connections of each case were linked to form the final analysis results.

## Results

### General information of study participants

There were nine study participants in this study, all of whom were six hospitalized patients with AA kindred transplantation and three medical staff working in hematopoietic stem cell transplantation wards and hematology departments. The average interview duration was 32 min. The general information of the patients is shown in Table 1, and the general information of the medical staff is shown in Table 2. The changes in the *Chinese version of the Posttraumatic Growth Inventory* (C-PTGI) scores of the patients are shown in Table 3 and Fig 1.

### PTG experience of patients with AA kindred transplantation

By coding and categorizing the interview data of the six patients, four themes were obtained: the period of agonizing, the period of ambivalence and struggle, the period of rooted growth, and the period of rebirth harvesting (see Fig 2).

**Theme 1: period of agonizing.**  The period of agonizing refered to the period from the diagnosis of AA to the time when the patient learned that he/she had to receive a transplant, firstly, he/she suffered from emotional fluctuations in the early stage

**Table 1. General information questionnaire for patients (n = 6).**

| Number | Sex | Age (years) | Marital status | Cultural level | Religious beliefs | Any current blood transfusions | Disease duration (months) | Date of Transplantation | Related Transplant Provider | Related Donor's Age (years) | Degree of HLA compatibility | Patient's blood type before related transplantation | Patient's blood type after parental transplantation | Type of transplantation |
|---|---|---|---|---|---|---|---|---|---|---|---|---|---|---|
| S1 | F | 30 | unmarried | High School | None | None | 96 | 14/8/2020 | Brother | 25 | 6/10 | B | B | 1,2,4 |
| S2 | M | 38 | married | College | None | None | 52 | 29/10/2020 | Son | 9 | 5/10 | A | A | 1,2,4 |
| S3 | F | 25 | married | Undergraduate | None | None | 122 | 24/5/2021 | Sister | 18 | 7/12 | O | O | 1,2,3,4 |
| S4 | M | 20 | unmarried | High School | None | None | 22 | 3/9/2020 | Sister | 29 | 5/10 | A | B | 1,2,4 |
| S5 | M | 52 | married | Junior High School | None | None | 13 | 21/4/2021 | Daughter | 25 | 6/12 | B | O | 1,4 |
| S6 | F | 35 | divorced | Junior High School | None | None | 7 | 15/10/2021 | Brother | 33 | 6/12 | A | A | 1,2,3,4 |

F: female M: male Transplant types: 1. bone marrow; 2. peripheral blood; 3. umbilical cord blood; 4. mesenchymal stem cells.

**Table 2. General information questionnaire for medical staff (n = 3).**

| Number | Sex | Age (years) | Marital status | Cultural level | Religious beliefs | Occupation | Years of work experience | Title |
|---|---|---|---|---|---|---|---|---|
| M1 | F | 32 | Single | Master | None | Nurse | 10 years | Supervisor Nurse |
| M2 | F | 27 | Single | Master | None | Nurse | 3 years | Junior Nurse |
| M3 | M | 49 | Married | Doctor | None | Doctor | 27 years | Chief Physician |

F: female M: male

**Table 3. Changes in C-PTGI scores of patients.**

| Number | C-PTGI-1 | | | C-PTGI-2 | | |
|---|---|---|---|---|---|---|
| | Scores | Level of C-PTGI | Duration of transplantation | Scores | Level of C-PTGI | Duration of transplantation |
| S1 | 82 | high | 11 months and 20 days | 72 | high | 18 months and 10 days |
| S2 | 57 | low | 9 months and 5 days | 57 | low | 12 months |
| S3 | 54 | low | 2 months and 25 days | 63 | medium | 5 months and 24 days |
| S4 | 78 | high | 12 months | 93 | high | 17 months and 25 days |
| S5 | 79 | high | 4 months and 15 days | 83 | high | 6 months and 10 days |
| S6 | 91 | high | 25 days | 92 | high | 4 months and 10 days |

C-PTGI: low level (0–59); medium level (60–66) ; high level (≥ 67)

of the disease when he/she found it difficult to accept the reality of cataract, and then he/she suffered from the agonizing pain of living with the disease in the course of the subsequent treatment, and then he/she had to choose transplantation due to the poor results of the conservative treatment, but because of the difficulty in obtaining an unrelated donor, patients with cataract had to start the long waiting for an unknown result. Overall, the agonizing period for patients with AA can be characterized by 2 aspects: "the fear and shock of the diagnosis of AA" and "the uncertainty of transplantation".

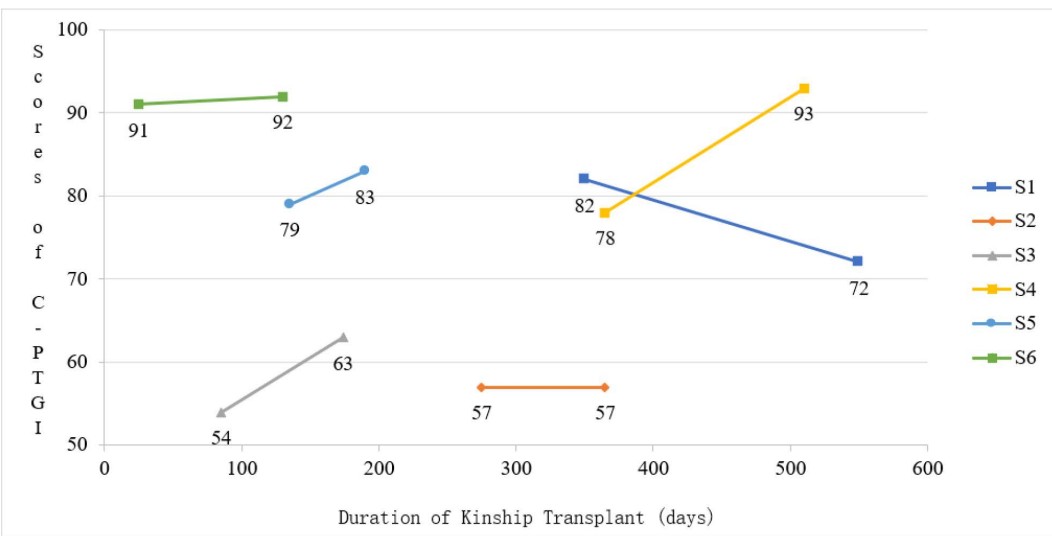

**Fig 1. Change of C-PTGI score of study participant.**

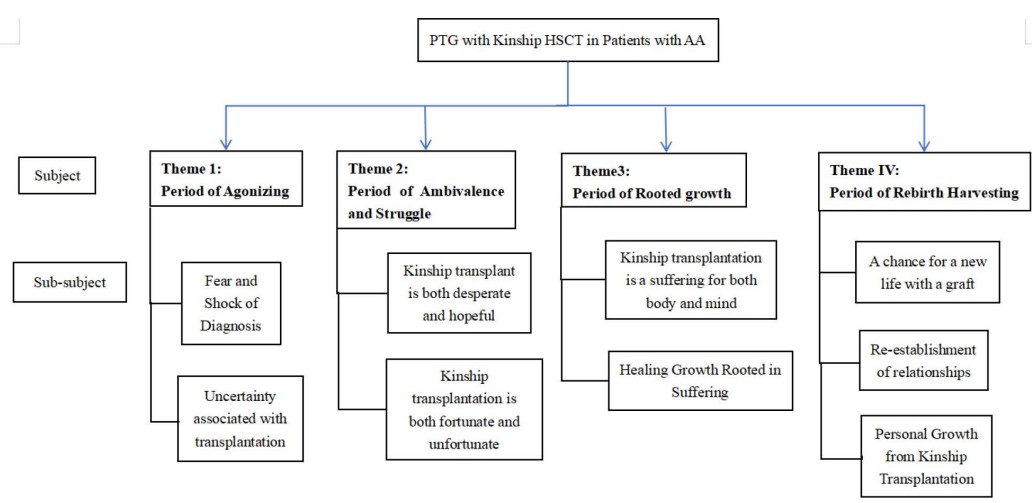

**Fig 2. Thematic framework diagram.**

① **Fear and Shock of Diagnosis:** Fear and Shock of a diagnosis of a disorder refers to the fact that the sudden catastrophe of an illness is a conflict and discomfort to the otherwise peaceful mind of a person with a reoccurring disorder, and that this unpredictable and life-threatening event often challenges the assumptions and beliefs about the world and the self and breeds a variety of negative reactions.

Often, when patients were confronted with a definitive diagnosis of a disease, their first reaction is shock, fear, and even denial. The suddenness of the diagnosis and subsequent treatment took the patient by surprise, and life no longer ran according to a pre-determined trajectory. Some patients were initially immersed in a shattered ego world at the time of diagnosis, and then fell into "diagnostic shock", struggling with questions such as, "Why do I have this disease? What should I do?" These questions were accompanied by a deep fear of death.

*When I was first diagnosed with AA I was especially afraid of dying young, the doctor told me that I might lose my life if I wasn't careful, my voice was trembling when I asked my teacher to take a leave of absence, I couldn't figure out how I could get remission when I was so young. (S4)*

*When I was diagnosed with AA I collapsed, my whole body collapsed, and I cried for hours at a time....... The doctor talked about transplantation, and while mentioning transplantation, he also talked about death, and as soon as a person hears about death, he will have fear. (S6)*

*When these patients were first diagnosed, they always asked us, 'Can this disease be cured?' 'I found it hard to accept that I had this disease, and I can't figured out why I suddenly got it.'(M1)*

Some patients were unable to accept the painful results of the diagnosis of disease, but when it was difficult to vent their negative emotions, they blamed their misfortune on fatalism, believing that the illness was a "man-made disaster", "heavenly calamity", and a predestined predestination, so as to rationalize the sudden onset of the disease. There was a rationalized explanation to divert the patient's resentment and condemnation, and to balance the pain in the heart. Although negative attribution styles such as fatalistic attribution could help kinship transplant patients to avoid inner pain for a short period of time, patients with negative attribution styles were more likely to suffer from depression and anxiety as well as PTSD in the long run.

*Before, I hated fate very much for being unfair, and it was a misfortune that made me get the (reoccurring) disease...... It might be the saying, 'It's a blessing, not a curse, it's a curse that can't be avoided'. (S6)*

In addition, most patients will first use conservative treatment after the diagnosis of AA. As the disease progressed, the patients gradually lost the hope for the future due to the lack of effectiveness after a long conservative treatment. The endless days was like a deep bottomless black hole without sunlight, so that the body and mind suffered from torment and torture.

*Suffering since 2017, has been taking medication, a little bit can not see the light, and the blood transfusion therapy makes me feel tired. Currently, the doctors don't have a relatively efficient treatment programs, just waiting passively. Take a step and take a look...... (S2)*

*I stayed at home for three or four years at that time, I couldn't go anywhere. But there are no peers in the community who have nothing to do all day, it's really hard for me, it feels like I can't see tomorrow at all in the dark night. (S1)*

In the process of conservative treatment, although some patients with AA also had a certain degree of family, social and other external support to properly regulate their emotions, they more often needed to endure the hardships brought by the disease and conservative treatment alone.

*Stress can not say to my parents, what is the use of saying those grievances, they are busy every day, and they can not do anything about it, because the disease is in me, and they are not a doctor, can not solve my problems, only to get money to see a doctor...... and friends are not in the same circle of life for a long time. They can't understand my pain. (S1)*

② **Uncertainty associated with transplantation:** The uncertainty associated with transplantation means that patients with AA who have to undergo transplantation when conservative treatment is unsuccessful or unavailable are uncertain about the process, the outcome, and the availability of an unrelated donor. Transplantation was the only "life-saving" option for patients with acute or severe AA, and even though the risks of transplantation are significant, there was no other choice for the patient.

*Before the transplant, I was waiting for death, but it was a matter of time, and the transplant operation was the only choice for my body, and I could only go this way. (S5)*

*Many patients didn't see much improvement with simple drug treatment, or in the acute stage of the disease, transplantation was the only way. Although they didn't know whether they can recover well after transplantation, they had no choice. (M3)*

For patients with chronic or non-severe AA, after undergoing conservative treatment for 1–2 months to 10 years or more, they had to face the "risky" choice of transplantation due to the poor efficacy of the treatment, which cannot be maintained.

*Doctors said that transplantation is the best option for this relapse, ten years ago we tried ATG, and then we have been on medication, so we can say that we have tried everything, so we can only transplant this time. (S3)*

While transplants were seen as treatment options for healthcare workers, they were a life-staking gamble for patients.

*Then taking this path is also a gamble, because transplantation is risky, if it is successful like this is the best, if it is not successful, I can't help. (S5)*

At the same time, the medical cost of transplant treatment was huge and cannot guarantee a "100% success rate", which brought emotional exhaustion and financial burden to patients.

*Because the doctor won't guarantee me, I have to bear this risk myself, I will definitely worry that I spent so much money to go down, and finally the person still walked away. (S2)*

*If the transplant is not good, the family will be broken, and the money will be spent and the people will be gone. (S5)*

Whether it was acute or severe aplastic patients, or chronic or non-severe aplastic patients, who were anxious and worried about the transplant process and outcome, it could be found that the uncertainty that comes with transplantation is a universal emotional experience.

*What I think about before transplantation is whether it can be achieved, and then I am not very clear about chemotherapy, and I am worried about whether it will be very uncomfortable and difficult, including rejection after transplantation or recurrence, etc., I will also consider more, after all, in the face of something I don't know, I am still a little worried. (S2)*

The deeper meaning of anxiety about "transplant failure" was the fear of death, and S6 even had written a testament to arrange for the aftermath in advance.

*My main fear is that the stem cell implantation will fail and then I will have to re-transplant again, and I am also afraid that there will be a rejection of stem cells between the patient and the donor, which will then lead to failure of the transplantation, and then I will gone. (S4)*

*At that time, I had my testament written, I really just wrote it while crying, but there was no way, this kind of thing had to be arranged, I was afraid of the eventuality (of death), so I had to explain the aftermath to my brother before the transplant. (S6)*

In the life-threatening, life suffered a serious injury, some of the preparatory transplant patients also experienced the cruelty of human indifference, which was a kind of betrayal for them "down the well", resulting in additional trauma "attached" to the disease.

*Now I know what it means to 'see true love in times of trouble', for example, my cousin, we grew up playing with a very good relationship, but this time I was sick, not to mention lending me money for transplantation, they did not even a word of concern.*

Generally, unrelated transplantation was the preferred option for patients with remission, but its main disadvantage was the scarcity and difficulty in obtaining unrelated donors. Most patients start a long and torturous wait for a 'small hope'. S6 stated that the probability of finding an unrelated donor was comparable to winning the lottery.

*If there is a stranger who is willing to donate, definitely I will choose the stranger, I would rather borrow more money and make up more money to him, but the probability of a stranger being able to match you is just too small, it's harder than winning the lottery. (S6)*

S4 believed that even after the time spent on the agonizing wait may ultimately be fruitless.

*If we consider the Chinese Bone Marrow Bank we have to wait torturously for months again and we may not be matched. (S4)*

In addition, some unrelated donors may regret donating before the transplant, and then only death awaits the patient with AA.

*Because some people in the bone marrow bank will regret donating before the transplant, which is more troublesome, just afraid that I have received chemotherapy, and the donor refuses to donate. Right, I will only have a dead end. (S3)*

And when patients with AA were waiting in vain for an unrelated donor, doctors often advised them to consider the ultimate option of a kindred transplant, because at this point a kindred transplant was the last chance.

**Theme 2: period of ambivalence and struggle.** It refered to the period from the time when the patient hesitated to make a decision on a graft to the time after the decision was finalized, and before the patient was admitted to the warehouse. Previously, patients with AA had difficulty in accepting the fact that they were sick and barely felt that they could live, but they were told that their condition had further deteriorated and that they had to save their lives through HSCT, and therefore they were caught in the dilemma of "survival at the cost of hurting their loved ones" and "death". Under the impact of related transplantation, the patient's re-established sense of security and self-identity was destroyed again, leading to more entangled and painful psychology, which lasted until the transplantation in the warehouse, and finally the patient had to recognize that related transplantation was the only hope to live and return to a normal life, and accept the treatment of related transplantation. The ambivalent struggling period of patients with AA mainly includes 2 aspects.

① **Kinship transplant was both desperate and hopeful:** Kinship transplantation was both desperate and hopeful for patients with AA. Desperate because waiting for a scarce unrelated donor was a dead end, but patients felt strong resistance to the idea of a related transplant as the only way out; compared to patients who cannot be matched with a suitable donor, a related transplant gave them hope for survival.

Often, when conservative treatment failed and a suitable unrelated donor cannot be found, patients with AA ended up considering a related transplant because it is easier to obtain a related donor than an unrelated donor. And, with the emotional tied of kinship, a kindred donor basically won't regret donating.

*For example, in case the transplant fails and has to be retransplanted, if it's with my sister's bone marrow, at least there won't be the concern of refusing to donate. (S4)*

However, unlike the general impression that people were more inclined to accept help from their family members, to our surprise, all the transplanted patients interviewed in this study reported that they preferred to receive a transplant from a 'stranger' (unrelated donor), even though a related transplant was the last option for them at that time.

*I've seen other people drawing peripheral blood in the drying room, I can't accept so much blood being drawn from my brother, it's really too dangerous, if I'm not careful, the person will be gone, it's really bloody and scary at the same time. (S1)*

*Many patients felt guilty and worried about their relatives' health when they saw a large bag of blood drawn from a blood donor during reinfusion. (M1)*

Even for those patients who were offered transplants from their "offspring" to "parents", they would rather gave up the treatment than accepted a transplant from their children, and they did not want to face the psychological trauma caused by related transplants, so they were strongly resistant to related transplants.

*Sometimes the thought of going for their son's bone marrow is really more painful than dying themselves, and that would really be better to die earlier. (S2)*

*Those patients who received donations from their children, especially those whose children were still in school and were relatively young, as parents, actually felt very painful in their hearts.(M2)*

At the same time, a kindred transplant was the last hope for a desperate patient with remodeling who had failed to match an unrelated donor. Some patients hoped to be cured of the disease and get rid of their patient status through kinship transplantation.

*I hope that through my sister's transplant I can break the root to reach a cured state, because taking medication for the rest of your life makes you feel like you are still a patient, and I want to get rid of that. (S3)*

*It was my brother's persistence in giving me a transplant and not giving up on me when I was sick that gave me hope to live. (S6)*

Especially for those patients who were matched with an unrelated donor successfully, but reversed their decision halfway, or those who failed to be matched with a related donor, or whose relatives were also unwilling to be matched to donate hematopoietic stem cells, the patients felt that they had been blessed with a lot of misfortune.

*My brother can match with me is already a desperate hope, before I heard that there is a patient who has been sick for more than ten years, his family is very rich, and the Chinese Marrow Bank has also matched several successfully, but then they all regretted it, and then his mom gave him a sister who still could not match, and that's the real despair. (S1)*

② **Kinship transplantation was both fortunate and unfortunate:** It was both fortunate and unfortunate that a patient with AA chose a close relative as a transplant donor. Fortunately, a kinship graft was the best treatment for a patient who was critically ill or in a time-critical situation, and could save the life. Unfortunately, the decision to go for a graft was a stressful one, and may result in a lifelong emotional indebtedness to the graft donor.

When a patient with AA was in severe condition or can't wait to be matched with a suitable donor, a related transplant was the most favorable and last chance for the patient to save his or her life.

*Many patients cannot waited for the matching of the Chinese bone marrow bank due to their urgent condition. The only way was to search for suitable bone marrow from relatives, and many relatives were willing to match patients to see if they can match. (M2)*

*Because at that time, (the Chinese bone marrow bank) hung out and did not respond for more than a month, and later had to have a blood transfusion every 4 days. The doctor said that it was always not good for blood transfusion, and it is not a good idea to delay it any longer, so it is most suitable for his son to transplant. (S2)*

There were also patients with AA who believed that a related transplant would have relatively less rejection and a faster physical recovery, which was one of the fortunate ones.

*I've heard them say that the cells grow a little bit faster with their own family's bone marrow, and that's probably the only benefit. (S2)*

Although patients with AA can also recognized that kinship transplantation was the most beneficial treatment for them, the objective fact that kinship transplantation was beneficial in contradiction with their subjective emotional experience of being unlucky. Some patients with AA believed that they had already burdened their families with their disease, and now they owed their families again by asking their relatives to provide blood stem cells and undergo even more physical pain, which was irreparable.

*Let my son donate bone marrow in my heart is both heartache and guilt, because he is only eight years old, too young to understand, to put it mildly, a child of the same age, he bears what he shouldn't bear. (S2)*

In addition, most of the patients with AA thought that the kinship transplantation was a gamble for an unknown result at the expense of the health of the related donor, for which they would be accompanied by endless sorrow and pain, which was the main reason for the patients' hesitation about the related transplantation. For example, S2 mentioned *worrying about my son* 16 times in the interview, and S5 mentioned *worrying about harming my daughter's health* 6 times in the interview. The majority of patients with AA were concerned that related donors would be more susceptible to disease, especially blood disorders, after transplantation.

*I will consider whether drawing my daughter's bone marrow and stem cells will harm her body, and I am also afraid that it will have a bad impact on her life in the future. I'm afraid that she may be anemia in the future, because of I don't know anything about the medicine."(S5)*

There were also patients who were afraid that their life expectancy would be shortened after the "life for life" of a related donor, so they were burdened with tremendous psychological pressure.

*I was thinking about whether the transplant would affect the length of his life, and I was worried that it would be my brother exchanging his ten years for my ten years, for example, he could live to be 90 years old, but after donating his bone marrow to me would he live 10 or 20 years less, and how would I make up for the less years? (S6)*

Meanwhile, if a patient with AA undergoes a related transplant, the family would have to bear the cost of transplantation for both the patient and the related donor, and the financial double burden would drag down the whole family, which would greatly decrease the quality of life.

The financial burden of disease is the main factor affecting the PTG level of patients, and patients with a high financial burden have a low PTG level. For example, in S5's interview, he mentioned 7 times that AA was a "catastrophic disease" that devastated his family and considered himself a family burden.

*I was really afraid of the cost of transplantation treatment, I needed money for transplantation, my daughter to give me blood donation also need money, essentially I was the backbone of the family, but now it brings catastrophic illness to my wife and children. We owed a large debt, how to ask my wife to pay. (S5)*

*Patients often experience significant mental stress, especially since they were the backbone of their family. After falling ill, the family's financial resources were almost non-existent, and the financial pressure could become immense. After relatives donated hematopoietic stem cells, they often needed a period of recuperation and care. This period may be a challenge for the entire family of the patient.(M3)*

**Theme3: period of rooted growth.** The period of rooted growth was the time between when a patient with AA entered the transplant warehouse and the completion of hematopoietic reconstruction, which generally took about a month. As time passed, patients with AA processed the traumatic event of receiving a graft on their own, developed a clearer understanding of grafting, and eventually accepted the reality that they need a graft to save their lives. After entering the transplant warehouse to receive grafting treatment, although the patients with AA were still suffering physically and mentally from their ordeal, they realized that life does not belong to them alone, but also to the loved ones who gave it to them. As a result, patients strived to find survival support from their suffering and gradually take control of their painful emotions, detached themselves from unrealistic assumptions, and strived to move toward growth in the constant battle between trauma and adjustment. The period of rooted growth for people with remodeling consists of 2 main aspects.

① **Kinship transplantation was a suffering for both body and mind:** After entering the transplantation warehouse, patients with AA had to endure the physiological side effects caused by drugs, and also had to bear the psychological burden caused by the trauma they had created for their family members, and sometimes the psychological pain in turn would further aggravate the physiological pain.

During the pretreatment period, patients with AA had to endure side effects such as vomiting, diarrhea, fever, mouth ulcers, etc. caused by chemotherapy drugs and other medications, and most of the patients thought that the pretreatment period after entering the warehouse was the most difficult time.

*The time growing cells, when I was vomiting and diarrhea and had a fever in the transplant warehouse was the most difficult. (S1)*

*After chemotherapy, the gastric mucosa broke, and the stomach pain was as excruciating as a knife twisting inside, so I couldn't sit or lie down, I was bloated and uncomfortable, and twice the pain was so severe that I couldn't sleep until the doctor dispensed medication in the latter part of the night. (S2)*

Prior to hematopoietic reconstruction, patients with AA had to endure severe bone and muscle soreness and pain, a pain that cannot even be relieved by painkillers. This was the extreme point of the patient's own physical pain; if the pain before was the pain of fear of not being able to be normal, the pain here was the real and deeply felt organic pain from the bone marrow.

*Then when the bone marrow started to grow, the bone pain in my back was so painful that it was almost (dead), and I was directly given to cry from the pain. (S1)*

Before the reconstruction of the hematopoietic system was completed, the blood picture of patients with AA often fluctuated greatly, and the mood of the patients and their families would be greatly affected by it.

*In the beginning, the hemogram was growing exceptionally well, and it just dawdled up, and then I don't know what the reason was, but the platelets dropped from 80,000 all the way down to 10,000 and my family was very worried. (S3)*

*Before the reconstruction of the hematopoietic system was completed, patients and their families were more prone to anxiety. They would be very concerned about the results of various indicators in blood tests.(M3)*

In addition, the harm caused to the kinship donor by the kinship transplant further aggravated the patient's psychological pain. Due to various constraints, patients with AA had to "sacrifice" the body of their closest relatives to fulfill their need to "live", but this harm to their family members was not caused by others but by themselves, as if they had to suck the blood of their closest ones in order to save themselves.

Relative donors needed to be injected with hematopoietic stem cell growth factor before the transplant to increase the number of hematopoietic stem cells in preparation for the subsequent extraction of hematopoietic stem cells. Relative donors would experience physiological discomfort during the stem cell mobilization process, which the patients grieved in their hearts.

*In order for my daughter's stem cells to be separated better, she got that (hematopoietic stem cell mobilizer) shot before the bone marrow was drawn, and then was a physical reaction. So she was all so uncomfortable, and she couldn't sleep, and it was hard for me to think about it. (S5)*

*My mom sent a few pictures of my brother when he was having his peripheral stem cells drawn, and the tube had to be inserted into his neck, which is such a fragile place. I couldn't even imagine seeing the pictures, it was scary and touching… and moving. (S1)*

In addition, in some cases, both expected and unanticipated circumstances occur during transplantation, and the imagined deficit to the related donor before transplantation evolved into concrete suffering. When the pain originally imagined as well as unanticipated by the related donor was actually presented in front of the patient's eyes, the remittance patient's internal self-blame was further deepened.

*When my sister started to help me draw bone marrow and peripheral blood, she experienced numbness and cramps, which we didn't expect before. Then once she went to the toilet by herself and fell in the middle of the road and cracked her mouth, and then fell again the next day, but by then I was already in the barn and couldn't take care of her, and was worried sick. (S3)*

S6 felt that her brother seemed to have exchanged his life for her own, leading to suicidal thoughts. The strong psychological fluctuations caused her to pause infusion due to high electrolyte levels that day. The doctor also informed her that her condition was critical. At this time, the physical and mental pain caused by the transplantation has far exceeded the pressure caused by the disease itself.

*My brother came up to see me after the bone marrow was drawn, the body was very weak being supported by my sister-in-law, I cried at that time. That night I felt that my brother had taken half a life in exchange for my life. Why do I have to live like this? My brother has to suffer for me, but also to pay so much money, thinking of these things my whole body is very crazy, cranky, and would like to die. Because of the stress caused my electrolytes to be too high that day the situation was particularly dangerous, the I.V. was all suspended and I almost couldn't make it through. (S6)*

② **Healing Growth Rooted in Suffering:** Rooted in Suffering refered to the fact that after entering the transplant warehouse, the patient was rooted in the suffering caused by the graft, and strived to draw strength from that suffering to break through the ground and ultimately make positive changes. Although the transplantation process involved a lot of pain and suffering, it was because of the transplantation that the patient gained the determination to live a good life.

After hematopoietic stem cell transfusion, patients with AA experienced severe pain during the process of rebuilding the hematopoietic system, but they knew that this agonizing physical pain was the "darkness before the dawn". At that time, the patients were happy despite their physical pain, and their pain tolerance was gradually strengthened during the process of grafting. PTG as a stage of mental journey was often viewed as a coping strategy or a coping effort in the face of persistent pain, and that pain was sometimes more easily tolerated by a person who realized the meaning and value of his or her experience. It was by discovering the value of the traumatic experience that the kindred transplant patients in this study were able to better coexist with their trauma.

> *More than half a month after transplantation people began to be sore and swollen, every other day white blood cells came up, not long after the hemoglobin also came up. Since it can grow, the body hard to endure. At least the soreness and swelling shows that the cells are starting to move inside, is a good sign. (S2)*

> *As long as the cells went up I didn't worry about it, including growing the cells that kind of pain into every bone crevice I can endure. It means that my brother's bone marrow has fully survived in my body, I don't have to worry about everything in vain or what, so I was especially happy at that time, the pain was also happy. (S6)*

> *During the process of cell growth, patients often felt whole body soreness, but when I told them that it was a manifestation of cell growth, they would be quite happy. Although it was painful, they believed it was a sign before the arrival of the rainbow. (M2)*

During hematopoietic reconstruction, some patients with AA had gained a survivor's perception of themselves through downward comparisons between patients or verbal encouragement from healthcare professionals, and this cognitive shift was a key step in the development of PTG [32]. Downward comparisons were often associated with constructive coping [33], and by comparing themselves to "people who were worse off," related transplant patients became more certain that they would recover better.

> *I recovered quite well after the transplant, and the chimerism with my brother's genetic piece is also good, and now all three lineages are up, almost within the normal range, and the doctor also said that I am the faster recovering one in the transplant case, and I can get up by myself to do some simple exercises now. (S6)*

And before the hematopoietic reconstruction reached the standard and stabilizes, each lineage might fluctuate in the process of hematopoietic reconstruction to varying degrees. Some patients thought that the fluctuation of blood counts was also a manifestation of hematopoietic reconstruction, just like repairing wounds, and that hematopoietic growth needed time to accumulate.

> *When the indicator goes up I'll post it in the WeChat family group, but if it falls off I won't say anything. My families will be set on the indicators. But in my mind, it's normal to go up and down in early days, to give time for these cells to grow (laughs). (S3)*

In addition, all six patients hoped that the graft can help them get rid of the physical "patient" status, and at the same time teard off the psychological "patient" label, so as to regain the sense of self-identity of a normal person. Many patients with AA were eager to get "freedom of eating" and "freedom of movement" through grafting.

> *Now that my son has saved my life, when I am discharged from the hospital, I won't have to be confined to my room every day, I can go out and play around. (S2)*

> *Since the transfusion of my daughter's marrow cells, I don't have the pressure of blood transfusion. My appetite is better now, I can go for a walk here and there. (S5)*

 

Some patients with AA believed that they can return to the family and society as a "normal people" after a graft. Others believed that after they were cured by a graft, their family members could return to normal life too.

*Because if my body can succeed through sister transplantation, I can do what I want to do, such as work normally, earn money, go to places I want to visit, eat seafood, and have a wedding with my husband. (S3)*

Despite the countless tormenting and painful moments throughout the period of kindred transplantation, patients with AA found support for survival through the concerted perseverance of the entire family, and the strength given by the family was the source of motivation for patients with remodeling to live. The family was the main source of personal self-strength. Family members could help patients developed positive attitudes and changes in crisis by providing them with instrumental and emotional resources to improve PTG levels [34].

*My biggest feeling is my mom. She brought me food every day, and she was especially haggard during that time. I can't let the people who helped me down, I have to give myself confidence to live well. Living is means that my parent's care is not wasted, and my brother's donation will not be wasted, it's all worth it, right? (S1)*

For S2, the "courageous and determined" attitude of the related donor was the main driving force that pushed him to set his mind right and actively participate in the treatment.

*My son didn't say anything when he had his blood drawn, and that's when I felt that he was so brave. I couldn't think too much about the bad side of things anymore. So that's why I also try to get well quickly, like my child is trying so hard, I also treat well and do what I need to do, don't always think about those useless things. (S2)*

S4 believed that kinship transplantation gave the opportunity to interpret the meaning of life, then it was necessary to take advantage of it to realize the value of life and took responsibility for the life of the kinship donor, the family, the community and even the country. It was the spiritual support of the family that pushes the patients to fight against suffering and find a sense of the meaning of life to survive. Studies [35] had shown that the multidimensional nature of the meaning of life can effectively improve the quality of patients' lives and helped them to face their illness and life with positive emotions. Patients completed their personal meaning through feelings, contemplation, and reminiscence as the sense of meaning of life increases PTG emerged and gradually increased.

*I felt that my life was not particularly important, but the meaning of life was particularly important. I can die but I had to die meaningfully. In case of failure, then I can only feel pity, there were still a lot of things had not been done, the meaning of life had not yet been interpreted completely. Subconsciously inside there would be something can do to the society. If I died at a young age, who would support my parents in the future, so I had to live well when I succeed. (S4)*

**Theme IV: period of rebirth harvesting.** The period was between the completion of hematopoietic reconstitution (usually one month post-transplant) and 18 months post-transplant in patients with remodeling of the hemopoiesis. Previously, the patient with AA had been put the "old self" to death, burying the stem cells that had no hematopoietic function. Now, with the hematopoietic stem cells from the related donor, he or she was resurrected from the dead and reborn as a new creation, bringing about personal and other changes and spiritual transformation. In this study, although the time of growth from suffering was different for each individual, the vast majority of patients with AA had risen to the challenge of facing the pain of grafting, breaking through themselves and rebuilding a new cognitive schema through self-reflection, gradually realizing the "growth" metamorphosis. At the same time, demonstrating the strength and unlimited potential of their inner being. The rejuvenation period of patients with remission mainly included 3 aspects.

① **A chance for a new life with a graft:** The opportunity for a new life brought about by grafting refered to the fact that patients with AA were given a new life after receiving grafting surgery. Although patients with AA felt pain because they hurt their family members by receiving a graft, they turned their grief and anger into strength by cognitively processing the traumatic event of grafting to obtain a "new me".

All 6 patients with AA mentioned that they were able to rebuild their lives for the second time through the graft, and in some cases, they were able to change their blood type.

*The (kindred) transplant would definitely be a continuation in my life, at least I don't have to stay up and wait for the Chinese Marrow Bank, uh, even though I gave him (my son) life, he gave me life in return, as if my son was here to save my life. (S2)*

*The (kindred) transplant was the rebirth of my own life with my daughter's blood, it's really as if she gave me life again. It was me gave birth to her, it turned into something like she gave birth to me. Now my blood type was the same as hers, and I had changed from blood type B to blood type O. (S5)*

*Many patients underwent personality changes after transplantation, becoming more like the donor's personality. So they themselves felt a sense of rebirth. (M1)*

In addition to the physical aspect, patients with AA of their philosophical outlook on life after the graft. The relative donor sacrificed a huge price to get the patient's current body, so they knew how to cherish time and life more, and they were more eager for a better life in the future. Some of the patients with AA expressed their appreciation of life after the relative transplantation. Studies [36] had shown that appreciation of life was a characteristic theme of PTG and that more appreciation of life promotes PTG emergence.

*Appreciation of life, to put it badly, this time it was a trip from the ghost gate, spend so much to heal. My daughter donated cells to me, my wife borrowed money everywhere. It meant that I have to cherish every day, to love my daughter and my wife, and all of my sibling family members. (S5)*

Other patients who had re-engineered their future work and life during their journey with trauma, believing that many new possibilities still exist in life. Studies [37] had shown that patients' post-traumatic search for new possibilities and increased hope for the future result in positive changes that facilitate coping with trauma. Despite not yet having returned to their families and society, patients with AA were able to actively plan their later life based on their situation. Such as S2 and S3, who planned to give back to their kinship donor's donation by way of financial compensation after the disease was cured, and who were filled with a constant stream of new motivation at the thought of their kinship donor's affectionate devotion.

*I had a few friends who were in technology, they planned to set up a company together when I got well. In short, no matter whether I found a good job or start my own business, at least I want to build up some wealth for my child. (S2)*

And, many patients with AA found it clearer and more practical to use their family members as a goal-oriented guide for making life plans. All the patients said that getting well first was the greatest reward for the kinship donor, and then they could afford to do other things they want to do.

*Now I was getting clearer about my goals. One was to work first after getting well, and then when I had money, I could take my kids out and travelled with my families. Only if I got well first can I go for the second step. (S2)*

At the same time, there were also many patients with AA who tried to overcome their bad habits after the related transplantation and paid more attention to self-protection and a healthy lifestyle. Since the lives of patients no longer belonged to themselves alone. Their lives were exchanged with the blood of their related donors, and their blood was flowing with the blood of those who loved them. So, spoil their own bodies was to spoil the true hearts and feelings of related donors.

*After receiving my brother's marrow, I paid more attention to correct bad habits, such as staying up late, waking up late, ah, well (thinking), did not eat spicy, and didn't do anything dangerous. I didn't want my brother to suffer in vain. (S1)*

*Now all aspects of life more regular. Get up early for morning running exercise, like before would sleep until noon. At night will not play games to the second half of the night, are on time to eat, go to bed at the right time. Smoking and drinking also all quit. After all, my life now is renewed by my son, I can't let his efforts go to waste. (S2)*

*Successful transplant patients would be very concerned about their future physical health. They would constantly consult us on how to eat, exercise, and other aspects, and their compliance with regular check ups was also relatively high. (M3)*

② **Re-establishment of relationships:**The reconstruction of relationship brought by grafting refers to the new establishment or change of relationship between patients with AA and others after grafting. Some patients were at home for a long time after illness, and the opportunity to communicate with their family members became more frequent. Because they were helped from all sides because of the kinship transplantation, their gratitude and empathy were greatly enhanced. These reconstructed positive cognitive schemas could also had a positive impact on the patient's condition.

All 6 patients with AA in this study believed that communication with family members became more frequent and family relationships became close and strong after the kindred transplantation. Wang et al. [38] showed that a cohesive family environment and cordial communication had a facilitating effect on PTG. Family members should maintain close companionship with patients, understand and meet their intrinsic needs, listen effectively and jointly feel the negative emotions of patients and provide psychological guidance, which helped their cognitive processing of traumatic stress and promotes their PTG.

*Before traveling outside, basically don't care family. The busiest time is two-thirds of one year are outside, can't take care of the family...... Now basically stay at home (laughs), after the disease is more concerned about the family. (S2)*

*During the stable period follow-up, patients and their families sometimes came to the hospital together. The whole family was quite concerned. I thought this process from illness to cure would increase their family cohesion. (M1)*

S4 was even more direct in removing the disconnect with the kinship donor.

*My sister and I would have a better relationship, after all, my body was flowing her blood. I used to feel disconnected because I didn't see much of her at school. Now I basically discuss everything with her...... being sick especially after the transplant and my dad became more communicative and closer because he used to work outside a lot. (S4)*

The patients with AA were able to get through the difficulties of the related transplant through the selfless dedication of their related donors and the careful care of their family members, so they decided to turn this deep-rooted kindness into action to repay their family members, and they would give all they had to their related donors to give thanks in return.

*I was very grateful to my sister for giving me the transplant, I would take more care of her in all aspects in the future. I wanted to return the favor and do more to help her, including returning more to my family as well. (S3)*

At the same time, most patients with AA had increased empathy for others after kinship transplantation, and some would change to a strong sense of altruism or prosocial behavior after increasing empathy. Studies had pointed out that "altruism after suffering" was a common experience in many traumatized patients, which were among the main manifestations of PTG [39–41]. Due to the commonality of kinship transplantation, many patients with AA and their families will communicate with each other and share their transplant experience. The patient's empathy ability would be improved in the process. This empathy was not just about sympathy for the suffering of others, but about a truly empathetic understanding.

*After the transplantation, I was more empathetic, I would think more about others, and care more about others. Especially if that person happened to be transplanted by a family member, I was willing to share everything I know with him. If conditions permitted, I would also give some financial help. Because I had experienced some pain, I would be more empathetic to the pain of others, and I would be more concerned about some things that had nothing to do with me. (S3)*

③ **Personal Growth from Kinship Transplantation:** Personal growth through grafting refers to the growth in personal self-relationships of patients with AA after grafting. Through processing and reflecting on the traumatic experience of the graft, patients with AA gradually gained self-identity and make timely psychological adjustments so that they can focus more on growth rather than the pain of loss.

Most of the patients with AA had a better ability to regulate and control their emotions after the grafting, and their psychological resilience increased. They were better able to face difficult situations positively, and the way of adjusting their mindset changed from a passive to an active one, and their positive and optimistic attitudes made them more resilient in the course of their illnesses.

*So I had to adjust myself in a hurry so that I had to get better quickly in all aspects, and my mindset had really changed a lot, I used to be more pessimistic, now I was more optimistic, and I believed I will be better in the future. (S6)*

Like S1 in the fight against the traumatic events of kinship transplantation, the degree of psychological tolerance for bad things wider and deeper, more able to withstand the "bad" end of the matter, with a kind of "in addition to life and death, were nothings," the lightness of mind.

*Now the mood was different, looking at what all felt good, felt very lucky. In the past, when the tempered film of my phone broke, I felt bad and thought I had to change the film again. But now I felt lucky that my phone was not broken, the screen was not shattered, and I can still use it, just that the film was broken. In general, I was less likely to take the bull by the horns now. (S1)*

Another example was S2, who no longer used negative coping styles such as avoidance to deal with things, but adopted "problem-solving-oriented thinking", accepted all the results openly, and tried to find a solution to cope with the difficulties in a positive way.

*There were some things that I can accept now, such as what the doctor told you about not thinking about what was going to happen and just accepting it and being prepared to deal with it. Something you didn't want to accept was impossible, like when I received my son's transplant, at the end of the day you still have to face it, you might as well just accept it and listen to the doctor on how to deal with it. (S2)*

In addition, after the grafting of relatives, the patients with AA gradually changed from "patients" to "normal people", began to find their own strengths and advantages, and fought against the trauma. Searching for themselves actively

and strengthening their ability to bear the pain during the days of coexisting with the trauma. After understanding the significance of their existence and suffering, they were inspired to turn pain into responsibility. Most patients with AA had resumed their family responsibilities after the kinship transplant, and their sense of self-worth and fulfillment has continued to increase.

*I can gradually took on the responsibilities as a father and a husband again. For example, the simple bad utensils at home can be repaired, and also spent time to accompany the children. (S2)*

## Discussion

### ① Guide patients with AA to have a correct understanding and reduce the uncertainty caused by genetic transplantation

In this study, after determining the treatment plan for grafting, patients with AA were unsure about the process of grafting, rejection, and prognosis, which resulted in negative psychology, such as *I will be worried if I don't understand the process of grafting* (S3), *If the grafting is unsuccessful, my daughter will suffer for nothing* (S5), and *It is too sinful to exchange my brother's life for mine* (S6), which affected the patients' quality of life. Therefore, it is necessary to guide patients with AA and their families to rationally understand the cognition of kinship transplantation, meet their information support needs, reduce the uncertainty caused by the traumatic event of kinship transplantation, and promote the development of PTG [42]. Medical staff should provide comprehensive personalized disease knowledge education to patients with AA according to different diagnosis and treatment periods. Guide patients in the early stages of diagnosis to distinguish the similarities and differences between diagnosis and other hematological diseases such as leukemia, and eliminate the concern of *"if it is leukemia, it's better to leave (S2)"*; Before entering the warehouse for transplantation, answer questions about the advantages and disadvantages of fully compatible and semi compatible transplantation, as well as the risks of transplantation to related donors, in order to alleviate the uncertainty caused by misconceptions such as *"older sister will often get sick after bone marrow extraction (S4)"* and *"younger brother will live ten years less after blood donation (S6)"* caused by related transplantation. Ultimately, guide patients to objectively and correctly treat their reproductive impairment and related transplantation, improve their sense of control over related transplantation, and promote the development of PTG.

### ② Building a stable support system to alleviate the trauma of patients undergoing genetic transplantation for recurrent disabilities

In this study, through interviews, we found that patients with AA have more obvious traumatic feelings under the stressful events of the related transplantation, which are mainly reflected in the fear of transplantation to harm the body of the related donor. The fear of transplantation to increase the financial burden, and the fear of transplantation to drag the family members in debt. Family and social support is closely related to patients' PTG [43]. Improving patients' family and social support can help them effectively alleviate the sense of disease trauma, enhance their confidence in actively participating in rehabilitation, and stimulate patients' PTG. Tehranineshat et al. [43] found that one's attribution of the cause of an event is an important factor influencing the reconstruction of the hypothetical world perception, and that a positive attributional style gives a clearer understanding of one's own emotional response to stressful events, which in turn promotes PTG. In this study, patients with AA were treated by healthcare professionals who *did their best to save their lives* (S5) and *careful care and comfort every day* (S6), as well as family members who *brought me meals every day* (S1), *called to encourage* (S3), and *buying me books to change my mood*. They gained confidence and courage to undergo a kinship transplant, and found support for survival from their families, which greatly buffered the trauma caused by the kinship transplant. In addition, the support from family members who *borrowed money from everywhere* (S4) and friends who *gave me money without saying a word*. Some patients with AA also received donations from strangers through the *Water Drop Fundraising*

*Donation* (S5), which enabled them to solve the economic crisis. This shows that a stable support system is the main way to effectively improve the trauma and PTG levels of patients with AA, which requires the collaboration of society, hospitals, and families.

③ **Exploring the potential of trauma repair and improving the sense of growth in patients undergoing genetic transplantation for the treatment of recurrent disabilities**

The study showed that the C-PTGI scores of four patients increased, one was flat, and one decreased after the kindred transplantation. The second scores of S3-S6 were higher than the previous one and all of them were in the middle-high level, which indicated that the patients with AA gained PTG through the kindred transplantation event. With the reconstruction of hematopoietic and immune systems, as well as the joint efforts of emotionally working together with family members, witnessing that the self-sacrificing of the kindred donor, the growth experience provided by patients is also becoming more and more abundant. S2, although he provided more experiences of growth after the grafting event in the interview, his scores were flat. Probably because his long-term experience of illness had made him comfortable with other things. And he was perhaps more conservative in his scoring; S1's second score was lower than the previous one, but still at a high level. Probably because his life was gradually getting on track after returning to society, and his emotions were slowly stabilizing. However, in the interview, we can appreciate his rich growth experience. Kindermann et al. [44] found that the patients who received the PTGI measurements would continue to think about the questions on the questionnaire for a period of time after the measurements were taken, and thus felt that they had gained a lot. In this study, the C-PTGI was used to measure patients prior to the interview to obtain the psychological state of the kindred transplant patients in real time, as well as to give the patients something to think about.

Although the point of research is not to force all patients to have access to PTG, they should believe that each kinship transplant patient has the potential to heal themselves and eventually grow. However, the goal of psychological rehabilitation should not be to create growth in the patient, but to gently promote the growth of the person with AA after discovering the growth or growth tendency [45].

## Conclusion

In this study, IPA was used to obtain the PTG experience through the coding and classification of the interview data of 6 aplastic patients. The PTG experience of AA kinship transplant patients was obtained, including the period of agonizing, the period of ambivalence and struggle, the period of rooted growth, and the period of rebirth harvesting.

### Limitations

Firstly, this study adopted a homogeneous small sample sampling strategy. However, due to geographical, temporal, and manpower limitations, the participants in this study were all from the hematology department of a tertiary hospital in Zhejiang Province. Secondly, during the interview process, the interview was interrupted due to the influence of real-life situations such as changes in the patient's condition or temporary medical orders. Therefore, this study conducted segmented interviews with some research participants at the same time point, which had a certain impact on the quality of the interviews.

### Supporting information

**S1 Fig.  Ethics review opinion.**
(DOCX)

**S2 Table.  Interview outline.**
(DOCX)

**S3 Table. Informed consent.**
(DOCX)

**S4 File. General information questionnaire.**
(DOCX)

**S5 File. Chinese version post-traumatic growth rating index.**
(DOCX)

**S6 Table. Preliminary annotations and analysis examples.**
(DOCX)

**S7 Table. Examples of basic themes.**
(DOCX)

**S8 Table. Example of theme analysis.**
(DOCX)

## Author contributions

**Conceptualization:** Min Xu, Yue Pan.

**Formal analysis:** Xinrui Huang, Yue Pan.

**Funding acquisition:** Min Xu.

**Investigation:** Yue Pan.

**Methodology:** Yue Pan, Ting Liu, Yan Xu.

**Project administration:** Min Xu.

**Resources:** Xinrui Huang, Yue Pan, Menghua Ye, Yan Xu.

**Software:** Yan Xu.

**Supervision:** Min Xu, Menghua Ye, Ting Liu, Yan Xu.

**Writing – original draft:** Xinrui Huang, Yue Pan.

**Writing – review & editing:** Xinrui Huang, Min Xu.

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
