## [Decision Letter · Decision Letter 0]

PONE-D-23-37733Post-traumatic growth experience with kinship hematopoietic stem cells transplantation in patients with aplastic anemia: A qualitative studyPLOS ONE

Dear Dr. Huang,

Thank you for submitting your manuscript to PLOS ONE. After careful consideration, we feel that it has merit but does not fully meet PLOS ONE’s publication criteria as it currently stands. Therefore, we invite you to submit a revised version of the manuscript that addresses the points raised during the review process.

We look forward to receiving your revised manuscript.

Kind regards,

Boshra Ismael Ahmed Arnout

Academic Editor

PLOS ONE

Journal Requirements:

Reviewers' comments:

Reviewer's Responses to Questions

**Comments to the Author**

1. Is the manuscript technically sound, and do the data support the conclusions?

Reviewer #1: Partly

Reviewer #2: Yes

2. Has the statistical analysis been performed appropriately and rigorously? 

Reviewer #1: Yes

Reviewer #2: N/A

3. Have the authors made all data underlying the findings in their manuscript fully available?

Reviewer #1: No

Reviewer #2: Yes

4. Is the manuscript presented in an intelligible fashion and written in standard English?

Reviewer #1: Yes

Reviewer #2: Yes

5. Review Comments to the Author

Reviewer #1: Please see PDF attachment for detailed reviewer comments to the authors.

Reviewer #2: Dear Author;

thank you for the good work, but your study based on patients feelings analysis, this may considered as a bias in the scientific researches. in the next study, please include high number of cases and analyze the collected data statistically, to avoid emotional bias. and i would like you to survey the opinion of care giver a ( nurses , doctors, others ..) as a care giver to patients undergo organ transplantation. it will be a complementary study.

your study have great impact on the patients awareness , medical care giver and community in general.

thanks

6. PLOS authors have the option to publish the peer review history of their article (what does this mean? ). If published, this will include your full peer review and any attached files.

**Do you want your identity to be public for this peer review?** For information about this choice, including consent withdrawal, please see our Privacy Policy .

Reviewer #1: No

Reviewer #2: **Yes: ** Dr. Luma Hassan Alwan Al Obaidy

---

## [Author Response · Author response to Decision Letter 1]

18 Jul 2024

Dear editorial,

I have made revisions to the article based on the opinions of the reviewers and uploaded the relevant files as requested in the email.

Best regards,

Xinrui Huang

---

## [Decision Letter · Decision Letter 1]

PONE-D-23-37733R1Post-traumatic growth experience with kinship hematopoietic stem cells transplantation in patients with aplastic anemia: A qualitative studyPLOS ONE

Dear Dr. Huang,

Thank you for submitting your manuscript to PLOS ONE. After careful consideration, we feel that it has merit but does not fully meet PLOS ONE’s publication criteria as it currently stands. Therefore, we invite you to submit a revised version of the manuscript that addresses the points raised during the review process.

We look forward to receiving your revised manuscript.

Kind regards,

Boshra A Arnout

Academic Editor

PLOS ONE

Reviewers' comments:

Reviewer's Responses to Questions

**Comments to the Author**

1. If the authors have adequately addressed your comments raised in a previous round of review and you feel that this manuscript is now acceptable for publication, you may indicate that here to bypass the “Comments to the Author” section, enter your conflict of interest statement in the “Confidential to Editor” section, and submit your "Accept" recommendation.

Reviewer #1: (No Response)

Reviewer #2: All comments have been addressed

2. Is the manuscript technically sound, and do the data support the conclusions?

Reviewer #1: Partly

Reviewer #2: Yes

3. Has the statistical analysis been performed appropriately and rigorously? 

Reviewer #1: I Don't Know

Reviewer #2: N/A

4. Have the authors made all data underlying the findings in their manuscript fully available?

Reviewer #1: No

Reviewer #2: Yes

5. Is the manuscript presented in an intelligible fashion and written in standard English?

Reviewer #1: Yes

Reviewer #2: Yes

6. Review Comments to the Author

**Reviewer #1: ** Thank you for an opportunity to re-review this manuscript. I can see that the authors have made some changes, however without a response letter with detailed information about the changes made it was not always clear if a comment had been addressed.

In my view my original comments, particularly around the data analysis, Results and Discussion sections have not yet been addressed, either via changes to the manuscript or via explanation from the authors as to why the change is not possible.

Data Analysis – this section reads more like a how to do IPA than what was done in present study

Table 1 – the information about participants presented here is very detailed, can authors please confirm this can’t be used to identify participants, particularly in conjunction with quotes in Results

Results

Results – write up of themes. Can authors clarify what was finding of the present research vs more general comments? it may just be writing style – would expect tense to be past tense, and findings situated in terms of the present study participants, vs more general what tends to happen.

Results appears to have discussion-style interpretation woven in e.g., line 238, line 420. There are some valuable observations made here as well as connections to existing research. I suggest removing this information from the Results section and working into the Discussion.

Discussion

Information here would likely have been better presented in context of the themes, rather than interpreting individual participants circumstances. In my view the Discussion needs re-writing with a greater focus of situating the study findings within the broader PTG post stem cell transplant literature. Bringing in some of the material currently in the Results is my suggestion for a starting point.

**Reviewer #2:**  to the authors,

thank you for your efforts to explain and make the require amendments. the study maybe helpful for the AA patients and general awareness of the effects of stem cells transplantation experiment.

7. PLOS authors have the option to publish the peer review history of their article (what does this mean? ). If published, this will include your full peer review and any attached files.

**Do you want your identity to be public for this peer review?** For information about this choice, including consent withdrawal, please see our Privacy Policy .

Reviewer #1: No

Reviewer #2: **Yes: ** Luma Hassan Alwan AlObaidy

---

## [Author Response · Author response to Decision Letter 2]

17 Nov 2024

Dear Reviewers:

I'm sorry that Reviewer 1 didn't see the response letter with detailed information about the changes made. But I am certain that I have uploaded the response letter as requested by the editorial department. This reply letter addresses each item the Reviewer 1 mentioned in your second revision suggestions. Thank you very much for your review!

Q1�Data Analysis – this section reads more like a how to do IPA than what was done in present study.

A1�The specific approach of this study has been indicated in the appendix at the end of the paragraph. For example, in the Preliminary Annotation and Critique section, the end of the paragraph indicates that the specific content about this step in this study is in Table S6.

Q2�Table 1 – the information about participants presented here is very detailed, can authors please confirm this can’t be used to identify participants, particularly in conjunction with quotes in Results.

A2�The patient information provided in Table 1 has been obtained with the patient's consent, and after discussion by the authors, it is unanimously agreed that the patient's privacy will not be disclosed.

Q3�Results – write up of themes. Can authors clarify what was finding of the present research vs more general comments? it may just be writing style – would expect tense to be past tense, and findings situated in terms of the present study participants, vs more general what tends to happen.

A3�The relevant modifications have been completed in the first revised draft. The modified content was to change the tense of the research conclusion of this study to the past tense, and the general conclusion to the present tense.

Q4�Results appears to have discussion-style interpretation woven in e.g., line 238, line 420. There are some valuable observations made here as well as connections to existing research. I suggest removing this information from the Results section and working into the Discussion.

A4�The proposed modification suggestion has already been modified during the first revision. We removed the original lines 238, lines 420 and so on from the results section and discuss them in the discussion section.

Q5�Discussion

Information here would likely have been better presented in context of the themes, rather than interpreting individual participants circumstances. In my view the Discussion needs re-writing with a greater focus of situating the study findings within the broader PTG post stem cell transplant literature. Bringing in some of the material currently in the Results is my suggestion for a starting point.

A5�The discussion section is divided into paragraphs based on the topic for writing. When discussing arguments, some conclusions about the researcher in the results may be used. Write the content that is not applicable in the Results into the Discussion based on the previous opinion.

---

## [Decision Letter · Decision Letter 2]

PONE-D-23-37733R2Post-traumatic growth experience with kinship hematopoietic stem cells transplantation in patients with aplastic anemia: A qualitative studyPLOS ONE

Dear Dr. Huang,

Thank you for submitting your manuscript to PLOS ONE. After careful consideration, we feel that it has merit but does not fully meet PLOS ONE’s publication criteria as it currently stands. Therefore, we invite you to submit a revised version of the manuscript that addresses the points raised during the review process. The manuscript has been evaluated by one reviewer, and their comments are available below. At this time, the following revisions are requested for clarity: - Please carefully review the overall manuscript to update the language and provide definitions as needed for specificity and clarity. Examples of unclear language and/or undefined terms include but are not limited to: "transplantation in the warehouse", "entered the transplant bin", "entering the barn," "period of rebirth harvesting."  

- Please provide a rationale for the inclusion of interviews with healthcare workers, as well as more information about the recruitment of healthcare workers in the Methods Section.

- Ther inclusion criteria includes "experienced positive changes brought by the event of a kinship transplantation" Please provide a defintion for a positive experience and specify how this was determined.

- The statement on lines 611 - 614 may be unsupported as written should be rephrased. If the statement represents a summary of patient responses please ensure it is supported by direct quotations.

- Please include a limitations section.

We look forward to receiving your revised manuscript.

Kind regards,

Vanessa Carels

Staff Editor

PLOS ONE

Journal Requirements:

Reviewers' comments:

Reviewer's Responses to Questions

**Comments to the Author**

1. If the authors have adequately addressed your comments raised in a previous round of review and you feel that this manuscript is now acceptable for publication, you may indicate that here to bypass the “Comments to the Author” section, enter your conflict of interest statement in the “Confidential to Editor” section, and submit your "Accept" recommendation.

Reviewer #2: All comments have been addressed

2. Is the manuscript technically sound, and do the data support the conclusions?

Reviewer #2: Yes

3. Has the statistical analysis been performed appropriately and rigorously? 

Reviewer #2: Yes

4. Have the authors made all data underlying the findings in their manuscript fully available?

Reviewer #2: Yes

5. Is the manuscript presented in an intelligible fashion and written in standard English?

Reviewer #2: Yes

6. Review Comments to the Author

Reviewer #2: Thank you for your efforts. No further comments are required. The authors explain their project main ideas.

7. PLOS authors have the option to publish the peer review history of their article (what does this mean? ). If published, this will include your full peer review and any attached files.

**Do you want your identity to be public for this peer review?** For information about this choice, including consent withdrawal, please see our Privacy Policy .

Reviewer #2: **Yes: ** Assisstant Professor Luma Hassan Alwan AL-Obaidy

---

## [Author Response · Author response to Decision Letter 3]

8 May 2025

Thank you to the reviewers for their valuable feedback on this study. After carefully reading the opinions of the experts, our research team members responded to each review comment and made corresponding modifications in the text. The following is the specific content.

1.Please carefully review the overall manuscript to update the language and provide definitions as needed for specificity and clarity. Examples of unclear language and/or undefined terms include but are not limited to: "transplantation in the warehouse", "entered the transplant bin", "entering the barn," "period of rebirth harvesting."  

Thank you for your valuable suggestion. Based on this opinion, we have re examined the language that may require further clarification of concepts throughout the entire text. For the first three phrases mentioned in the revision suggestions, we believe that the main reason for the unclear definition may be the inconsistent use of words. Therefore, we will unify 'bin' and 'barn' in the text as 'warehouse'. However, the term 'period of rebirth harvesting' is a summary based on qualitative interviews, and the explanation for this period has been elaborated in detail in the following article.  

2.Please provide a rationale for the inclusion of interviews with healthcare workers, as well as more information about the recruitment of healthcare workers in the Methods Section.

Thank you for your valuable suggestion. Interviewing this group of people was based on the opinions provided by the previous round of reviewers. After discussion, we believe that healthcare workers, as a group of people who closely connect with patients during transplantation treatment, may have a direct understanding of the patient's emotional journey during this special period. In order to gain a more comprehensive understanding of patients' psychological changes, we have decided to conduct additional interviews with healthcare workers as a group. We have explained this opinion in the article and supplemented the inclusion and exclusion criteria for healthcare workers.

3.Ther inclusion criteria includes "experienced positive changes brought by the event of a kinship transplantation" Please provide a defintion for a positive experience and specify how this was determined.

Thank you for your valuable suggestion. In this study, we included a population with positive experiences, which mainly refer to positive experiences of post-traumatic growth. This definition has been presented in the introduction section.

4.The statement on lines 611 - 614 may be unsupported as written should be rephrased. If the statement represents a summary of patient responses please ensure it is supported by direct quotations.

Thank you for your valuable suggestion. This section is based on the patient's true oral account, rather than references.

5.Please include a limitations section.

Thank you for your valuable suggestion. This section has been added to the main text.

---

## [Decision Letter · Decision Letter 3]

Post-traumatic growth experience with kinship hematopoietic stem cells transplantation in patients with aplastic anemia: A qualitative study

PONE-D-23-37733R3

Dear Dr. Min Xu

We’re pleased to inform you that your manuscript has been judged scientifically suitable for publication and will be formally accepted for publication once it meets all outstanding technical requirements.

Kind regards,

Nusrat Saba

Academic Editor

PLOS ONE

Additional Editor Comments (optional):

Accept

Reviewers' comments:

Reviewer's Responses to Questions

**Comments to the Author**

1. If the authors have adequately addressed your comments raised in a previous round of review and you feel that this manuscript is now acceptable for publication, you may indicate that here to bypass the “Comments to the Author” section, enter your conflict of interest statement in the “Confidential to Editor” section, and submit your "Accept" recommendation.

Reviewer #2: All comments have been addressed

2. Is the manuscript technically sound, and do the data support the conclusions?

Reviewer #2: Yes

3. Has the statistical analysis been performed appropriately and rigorously? 

Reviewer #2: Yes

4. Have the authors made all data underlying the findings in their manuscript fully available?

Reviewer #2: Yes

5. Is the manuscript presented in an intelligible fashion and written in standard English?

Reviewer #2: Yes

6. Review Comments to the Author

Reviewer #2: Dear Editors,

The authors had explained the purpose of their research, made a linguistic correction, and added the medical staff's opinion, so I think it's a good manuscript to publish.

thanks

7. PLOS authors have the option to publish the peer review history of their article (what does this mean? ). If published, this will include your full peer review and any attached files.

**Do you want your identity to be public for this peer review?** For information about this choice, including consent withdrawal, please see our Privacy Policy .

Reviewer #2: No

---

## [Editor Report · Acceptance letter]

PONE-D-23-37733R3

PLOS ONE

Dear Dr. Xu,

I'm pleased to inform you that your manuscript has been deemed suitable for publication in PLOS ONE. Congratulations! Your manuscript is now being handed over to our production team.

Kind regards,

on behalf of

Dr. Nusrat Saba

Academic Editor

PLOS ONE